# Evaluation of Six Satellite Precipitation Products over the Chinese Mainland

Zhenwei Liu [1], Zhenhua Di [1,*], Peihua Qin [2], Shenglei Zhang [3] and Qian Ma [4]

1    State Key Laboratory of Earth Surface Processes and Resource Ecology, Faculty of Geographical Science, Beijing Normal University, Beijing 100875, China
2    State Key Laboratory of Numerical Modeling for Atmospheric Sciences and Geophysical Fluid Dynamics (LASG), Institute of Atmospheric Physics, Chinese Academy of Sciences, Beijing 100029, China
3    State Key Laboratory of Remote Sensing Science, Aerospace Information Research Institute, Chinese Academy of Sciences, Beijing 100101, China
4    College of Global Change and Earth System Science, Beijing Normal University, Beijing 100875, China
*    Correspondence: zhdi@bnu.edu.cn; Tel.: +86-10-5880-4622

**Abstract:** Satellite precipitation products have been applied to many research fields due to their high spatial and temporal resolution. However, satellite inversion of precipitation is indirect, and different inversion algorithms limit the accuracy of the measurement results, which leads to great uncertainty. Therefore, it is of great significance to quantify and record the error characteristics of different satellite precipitation products for their better application in hydrology and other research fields. In this study, based on CN05.1, which is a set of site–based interpolation data, we evaluated the accuracies of the six satellite precipitation datasets (IMERG–E, IMERG–L, IMERG–F, GSMaP, CMORPH, and PERSIANN–CDR) at different temporal scales (daily, monthly, and yearly) in mainland China for the period from 2001 to 2015. The results were as follows: (1) In terms of mean precipitation, IMERG–F was superior to other data in all areas. IMERG products and PERANN–CDR performed better than other products at all scales and were more suitable for precipitation research in mainland China. Site correction can effectively improve the accuracy of product inversion, so IMERG–F was significantly better than IMERG–E and IMERG–L. (2) Except PERSIANN–CDR, all precipitation products underestimated precipitation in the range of 1–4 mm/day and had a high coincidence with CN05.1 in the range of 4–128 mm/day. (3) The performance of six types of satellite precipitation products in summer was better than that in winter. However, the error was larger in seasons with more precipitation. (4) In the Qinghai–Tibet Plateau, where there are few stations, the inversion of precipitation by satellite products is closer to the actual situation, which is noteworthy. These results help users understand the characteristics of these products and improve algorithms for future algorithm developers.

**Keywords:** satellite precipitation products evaluation; Chinese mainland; IMERG; GSMaP; CMORPH; PERSIANN–CDR




## 1. Introduction

Precipitation is one of the crucial meteorological variables in research fields such as drought assessment and detection [1], flood forecasting [2], soil characteristic analysis [3], and hydrometeorological models [4], and its accurate measurement helps to conduct more comprehensive studies of hydrological and meteorological events. However, precipitation has large variability in temporal and spatial domains and often presents nonnormal distribution, so accurate measurements of precipitation have been a problem [5,6].

The traditional methods of measuring precipitation mainly include rain gauges and ground–based radars. The rain gauge can provide direct and accurate precipitation observations, but its spatial representation is relatively limited, especially for limited–data areas. Radar can provide precipitation information over a range of several hundred kilometers and thus is mainly suitable for measuring local precipitation events.

In contrast to conventional observations, satellite precipitation retrieval products have wider coverage and higher timeliness and can make up for the deficiencies in the spatiotemporal representativeness of precipitation gauges and ground–based radar. According to the types of sensors mounted on the satellite, the retrieval algorithms are mainly divided into four types: visible/infrared (VIS/IR), passive microwave (PMW), active microwave, and multisensory combination. The VIS/IR algorithm is the first proposed and simplest method. It estimates the precipitation by cloud top information measured in the VIS/IR wave band. The precipitation retrieved by the VIS/IR algorithm shows a continuous and accurate spatial distribution, but it tends to overestimate precipitation [7,8]. The PWM retrieval algorithm has more direct physical inference than the VIS/IR algorithm. Microwaves have a strong penetration of clouds and rain in the atmosphere, so they can obtain precipitation structure information in addition to the precipitation amount, which is different from the VIS/IR algorithm [9]. However, most of the PWM algorithms are optimized according to their respective sensor characteristics, inducing large uncertainty in the retrieval results [10].

The active microwave referring to precipitation radar (PR) retrieval algorithm is used to estimate the precipitation intensity and precipitation amount based on radar echo strength [11]. It not only absorbs the advantage of the ground–based radar algorithm but also monitors larger areas by satellite movement. Therefore, it has a high precipitation retrieval accuracy. Even so, the algorithm still has uncertainty and is mainly affected by the identification of some parameters related to the phase state of precipitation particles, raindrop spectrum, precipitation temperature, and precipitation inhomogeneity in radar imaging pixels [12]. The multisensory joint retrieval algorithm absorbs the advantages of different sensor retrieval algorithms and thus could obtain higher quality precipitation products. For instance, the Climate Prediction Center morphing (CMORPH) algorithm uses the infrared data of geostationary satellites every half an hour to interpolate PWM inversion data to obtain relatively fine spatiotemporal precipitation data [13]; the Tropical Rainfall Measuring Mission (TRMM) algorithm combines passive and active microwave data to improve retrieval ability for clouds and precipitation [14].

There are many precipitation retrieval algorithms, but they all have uncertainty. Therefore, the World Meteorological Organization (WMO) initiated a Program to Evaluate High Resolution Precipitation Products (PEHRPP) [15,16], aiming to promote the accuracy of satellite precipitation retrieval products from countries all over the world [17–20]. Note also that China joined the program at the 6th International Precipitation Working Group (IPWG) meeting in October 2012. Therefore, various satellite precipitation products have been evaluated separately in different studies in many countries. For instance, TRMM precipitation data performed better than CMORPH and PERSIANN in Nepal [21], North Brazil [22], and Africa [23]; however, CMOPRH outperformed PERSIANN in the United States [24]. In addition, the IMERG precipitation product is closer to ground station observations than TRMM in Singapore [25], and the TRMM product performed better than the Climate Hazards Group Infrared Precipitation with Stations (CHIRPS) in the lower Mekong River Basin of Southeast Asia [26]. It can be inferred that the accuracy of IMERG is higher than CHIRPS; however, an opposite situation, in which CHIRPS was superior to IMERG, occurred in the southwestern Pacific region [27]. Therefore, satellite precipitation retrieval products have different performances in different countries and regions.

There are also many studies on the evaluation of satellite precipitation products in China. For instance, Lu and Yong [28] evaluated the precipitation products of IMERG version 5 and GSMaP version 7 over the Tibetan Plateau; Zhang et al. [29] evaluated the hydrologic utilization of the TRMM and GPM precipitation products in a humid basin of China; Zhang et al. [30] provided a comprehensive evaluation of the accuracies of TMPA and PERSIANN in a semihumid region in northeastern China; Guo et al. [31] evaluated the PERSIANN–CDR, CHIRPS, and MSWEP for obtaining the characteristics of drought events in northwest China. Li et al. [32] found that although the corrected IMERG–F was more accurate than the uncorrected IMERG–E and IMERG–L, the correction exacerbated

the overestimation of precipitation. Liu et al. [33] evaluated TRMM3B42, CMORPH, and PERSIANN in the Poyang Lake basin using 52 rain gauge stations. Bai et al. [34] analyzed the temporal and spatial variation characteristics of long–term climate disaster cluster Infrared Precipitation Satellite (CHIRPS) QPEs in mainland China. Chen et al. [35] analyzed the structural errors of IMERG monthly precipitation products in mainland China from March 2014 to February 2015. Obviously, these evaluations were conducted in a certain region, and the products used to be compared were not uniform; thus, different conclusions existed. Notably, some studies focused on the evaluation of satellite precipitation products in China, especially for mainland China. However, most of them either assessed the applicability of a single satellite product in mainland China [36–38], lacking a comparison of different product performances, or two or three products over a shorter period, usually one to three years [39–43].

In this study, six commonly used satellite precipitation products were evaluated in mainland China from 2001 to 2015. The paper is organized as follows. Section 1 gives the introduction; Section 2 introduces the precipitation analysis dataset, the satellite precipitation products, and evaluation metrics; the evaluation results are demonstrated in Section 3; Section 4 discusses the possible reasons for the product difference; and the conclusions are presented in Section 5.

## 2. Materials and Methods

### 2.1. Ground Interpolation Product Production

The daily gridded precipitation data CN05.1 were used to evaluate the performance of the six satellite precipitation products. The CN05.1 dataset was produced by observations by the National Climate Center [44], China Meteorological Administration. It included more than 2400 national stations (basic, baseline, and general stations) from the China Meteorological Administration (Figure 1) and interpolated them as a grid dataset by thin–plate smoothing spline technology. The dataset ranged for all of China from 1961 to 2020. The dataset included precipitation and mean, maximum, and minimum temperatures. It not only had a high spatial resolution of 0.25° but also represented the actual change in meteorological variables relatively accurately [45]. Therefore, the CN05.1 data were used as a reference for comparisons of satellite retrieval precipitation products.

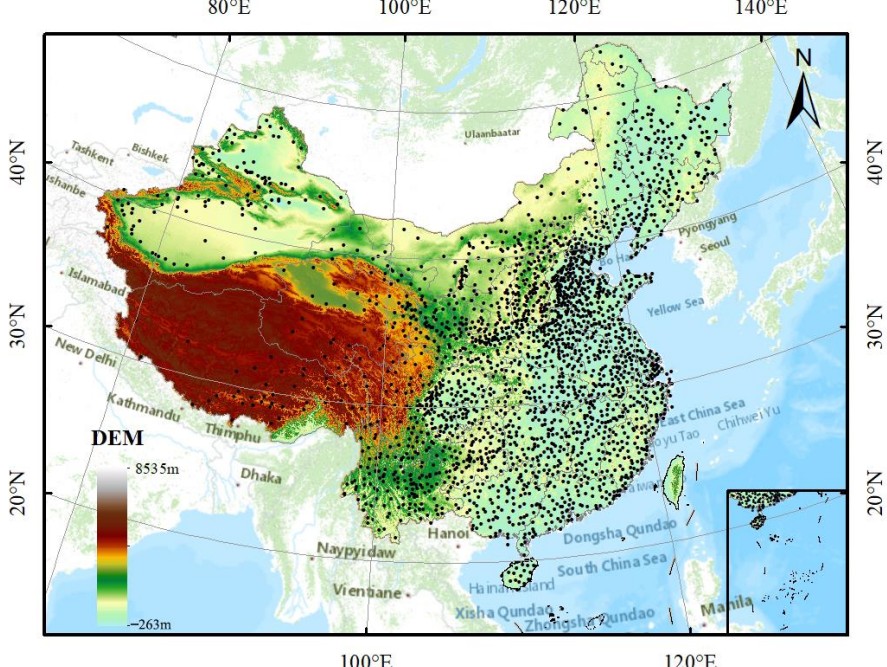

**Figure 1.** Distribution of rain gauge stations.

### 2.2. Satellite Precipitation Products

In this study, six satellite precipitation products were employed: Integrated Multi–satellite Retrievals for Global Precipitation Measurement (IMERG–E, IMERG–L, and IMERG–F), Global Satellite Mapping of Precipitation (GSMaP), NOAA's Climate Prediction Center morphing technique (CMORPH), and Precipitation Estimation from Remotely Sensed Information using Artificial Neural Networks–Climate Data Record (PERSIANN–CDR) (see Table 1). The satellite precipitation products selected in this study are not completely independent from each other: different products may use the same sensor, and one product may be incorporated into another.

**Table 1.** Summary of six satellite precipitation products in this study.

| Name | Resolution | Coverage | Data Source |
| --- | --- | --- | --- |
| IMERG–E | 0.1°/Daily | 60°N~60°S | 2000 to present |
| IMERG–L | 0.1°/Daily | 60°N~60°S | 2000 to present |
| IMERG–F | 0.1°/Daily | 60°N~60°S | 2000 to 2021 |
| GSMaP | 0.1°/Daily | 60°N~60°S | 2000 to present |
| CMORPH | 0.25°/Daily | 60°N~60°S | 1998 to 2019 |
| PERSIANN–CDR | 0.25°/Daily | 60°N~60°S | 1983 to present |

Compared with the TRMM, GPM increased the first spaceborne Dual–Frequency Precipitation Radar (DPR) and the Conical–scanning Microwave Imager/Sounder (CMIS). DPR is able to observe the internal structure of storms under and within clouds and CMIS has the ability to measure the size, type, and intensity of precipitation [46]. The IMERG algorithm integrates satellite retrieval from TMPA, CMORPH, and PERSIANN [47]. Precipitation estimated from PMW sensors was first calculated by the Goddard Profiling Algorithm (GPROF2017). After gridding and interactive calibration, the GPM Combined Ku Radar–Radiometer Algorithm (CORRA) product was then combined to a half hour $0.1° \times 0.1°$ (about $10 \times 10$ km) field [48]. Then, half–hour scale data were input into Climate Prediction Center (CPC) Morphing–Kalman Filter (CMORPH–KF), Lagrangian time interpolation scheme, and the Precipitation Estimation from Remotely Sensed Information using Artificial Neural Networks Cloud Classification System (PERSIANN–CCS) recalibration scheme. Finally, a bias correction was conducted using the monthly Global Precipitation Climatology Centre (GPCC) product to improve the accuracy of the product [47]. The IMERG precipitation products include three modes: early–run (IMERG–E), late–run (IMERG–L), and final–run (IMERG–F), with a spatial resolution of $0.1° \times 0.1°$ and coverage of 60°N–60°S. IMERG–E and IMERG–L are quasi–real–time precipitation estimation products which are released approximately 4 h and 12 h later, respectively. IMERG–E uses only forward deduction technology and is mainly used for disaster analysis and other adjacent forecasting applications. IMERG–L has added backward extrapolation, using richer data for day–by–day or longer analyses. IMERG–F was released after a delay of approximately 2.5 months and was corrected by precipitation analysis from monthly ground interpolation product stations, mainly for scientific research. In this study, the bilinear interpolation technique was used to convert data from 0.1° to 0.25°. The IMERG product was downloaded from NASA's Earth Observing System Data and Information System (EOSDIS) website https://disc.gsfc.nasa.gov/ (accessed on 21 August 2022).

As one of the Japanese GPM projects, GSMaP is a satellite precipitation product measured by the thermal IR and microwave radiometer (TIR–MWR) with a spatial resolution of 0.1° and coverage of 60°N to 60°S. The sensors associated with it include the IR sensor by the CPC (Climate Prediction Center) of NOAA, TRMM Microwave Imager (TMI) by TRMM, Advanced Microwave Scanning Radiometer for EOS (ASMR–E) by Aqua, and SSMI (Special Sensor Microwave/Imager) by the Defense Meteorological Satellite Program (DSMP) [49]. GSMaP comprises three variants: Rainfall Watch (GSMaP_RNL, GSMaP_MVK, GSMaP_NRT), GSMaP Realtime (GSMaP_NOW), and RIKEN Nowcast (GSMaP_RNC) [50]. The near–real–time data product GSMaP_NRT adopts the forward cloud vector motion scheme in

the processing process, while the standard product GSMaP_MVK adopts the bidirectional (forward and backward) cloud vector motion scheme. This study used GSMaP_RNL, which has the same algorithm as GSMaP_MVK (moving vector with a Kalman filter to estimate the precipitation rate). The GSMaP_RNL also used Japanese 55–year reanalysis (JRA–55) data as ancillary data to produce a homogenous and continuous dataset [51]. The GSMaP product was downloaded from the JAXA Global Rainfall Watch website https://sharaku.eorc.jaxa.jp/GSMaP/index.htm (accessed on 30 October 2021).

CMORPH is a new technology developed by the Climate Prediction Center (CPC) of the United States National Oceanic and Atmospheric Administration (NOAA). The primary characteristic is the integration of the IR data of geostationary satellites and MW data of polar satellites for the estimation of precipitation. The CMORPH algorithm makes use of the advantages of MW precipitation data and IR data, and uses IR precipitation data to calculate the cloud displacement of two time periods of MW precipitation data which is spliced into the global precipitation data. Finally, the precipitation calculation is completed by using time weighting [52]. The CMORPH satellite precipitation estimates are created in two steps. Firstly, the satellite–based global precipitation field is constructed. Bias in these integrated satellite precipitation estimates is then removed through comparison against CPC daily gauge analysis over land and adjustment against the Global Precipitation Climatology Project (GPCP) merged analysis of pentad precipitation over ocean. The biased–corrected CMORPH data are an 8 km × 8 km grid from 60°S to 60°N, starting on 1 January 1998. Due to the need for manual generation, CMORPH version1 (hereinafter referred to as CMORPH) used in this study is generally available with a delay of 3–4 months. The data used in this study were generated based on 30 min of 8 km data, with a spatial resolution of 0.25° and a temporal resolution of 1 day. The CMORPH datasets are available at ftp://ftp.cpc.ncep.noaa.gov/precip (accessed on 15 October 2021).

PERSIANN–CDR is developed and used by the Center for Hydrometeorology and Remote Sensing (CHRS) at the University of California, Irvine (UCI). The time resolution of this product is 1 day, and the spatial resolution is 0.25°, covering the 60°N–60°S from 1983 to present. PERSIANN–CDR is mainly used to study the trend of extreme precipitation events. PERSIANN algorithm is used to process GridSat–B1 infrared data. The Global Precipitation Climatology Project (GPCP) monthly product is used to adjust the PERSIANN–CDR. The PERSIANN–CDR product was downloaded from http://www.chrsdata.eng.uci.edu/ (accessed on 3 August 2022).

### 2.3. Metrics for Precipitation Product Quality

In this study, the root mean square root error (RMSE), mean error (ME), correlation coefficient (CC), and the Kling–Gupta efficiency (KGE) were used to quantitatively evaluate the accuracy of precipitation intensity. The probability of detection (POD) and false–alarm ratio (FAR) were used to evaluate the hit ratio of the precipitation product to the actual precipitation event. The KGE values range from $(-\infty, 1]$; the closer the KGE value is to 1, the better the agreement is between the modified product and the observed value [53]. POD represents the hit ratio of the estimate value, with values closer to 1 indicating more accuracy for the satellite production. The lower the FAR value is, the more accurate the estimate. These statistical indices have been widely used in previous studies [54,55]. ME, RMSE, CC, KGE, POD, and FAR are calculated using Equations (1)–(8).

$$\text{RMSE} = \sqrt{\frac{1}{N}\sum_{i=1}^{N}(Y_i - O_i)^2} \tag{1}$$

$$\text{ME} = \frac{1}{N}\sum_{i=1}^{N}(Y_i - O_i) \tag{2}$$

$$CC = \frac{\sum\limits_{i=1}^{N} (Y_i - \overline{Y})(O_i - \overline{O})}{\sqrt{\sum\limits_{i=1}^{N} (Y_i - \overline{Y})^2} \sqrt{\sum\limits_{i=1}^{N} (O_i - \overline{O})^2}} \tag{3}$$

where $Y_i$ is the estimated value at grid cell $i$, $O_i$ is its actual observation value (i.e., CN05.1 value), and $N$ is the number of grid cells.

$$KGE = 1 - \sqrt{(R-1)^2 + (\beta - 1)^2 + (\gamma - 1)^2} \tag{4}$$

$$\beta = \frac{\mu_s}{\mu_o} \tag{5}$$

$$\gamma = \frac{CV_s}{CV_o} \tag{6}$$

where $R$ is the linear correlation coefficient, $\mu$ is the mean precipitation, $\beta$ is the bias ratio, $\gamma$ is the variability ratio, $CV$ is the coefficient of variation, $s$ is the satellite precipitation value, and $o$ is the observation precipitation values.

$$POD = \frac{hits}{hits + misses} \tag{7}$$

$$FAR = \frac{false\ alarms}{hits\ +\ false\ alarms} \tag{8}$$

Hits (misses) indicate the satellite precipitation products that detect (miss) actual precipitation events captured by CN05.1. False alarms mean that the satellite precipitation products detect fake precipitation events. The contingency table for categorical verification statistics and the detailed definitions are shown in Table 2.

**Table 2.** Contingency table for comparing ground interpolation product and satellite precipitation products.

|  | CN05.1 ≥ Threshold | CN05.1 < Threshold |
|---|---|---|
| Satellite ≥ threshold | Hits | False alarms |
| Satellite < threshold | Misses | Correct negatives |

## 3. Results

### 3.1. Daily Precipitation

In this study, the CN05.1 dataset was used as the benchmark to evaluate the performance of different satellite precipitation products over Mainland China from 2001 to 2015. Table 3 presents the statistical results for six daily satellite precipitation products. The RMSEs of the six datasets ranged from 5.47 to 8.30 mm/day, and PERSIANN–CDR has the lowest RMSE value (5.47 mm/day) followed by IMERG–F (5.63 mm/day). The ME of IMERG–F was the lowest in the six products, and its slightly positive bias (0.01 mm/day) demonstrated that it had better consistency with the ground interpolation product. Significant bias occurred in CMORPH. Compared with IMERG–E and IMERG–L, IMERG–F had a smaller negative bias. This is because IMERG–F experienced site correction based on the GPCC precipitation gauge [56].

**Table 3.** Daily precipitation statistics of the six satellite precipitation products over mainland China from 2001 to 2015 *.

|  | RMSE (mm/Day) | ME (mm/Day) | CC | KGE | POD | FAR |
|---|---|---|---|---|---|---|
| IMERG–E | 5.73 | 0.07 | 0.49 | 0.46 | **0.74** | 0.25 |
| IMERG–L | 5.89 | 0.07 | 0.49 | 0.45 | **0.74** | 0.25 |
| IMERG–F | 5.63 | **0.01** | **0.5** | **0.48** | **0.74** | 0.25 |
| GSMaP | 7.45 | −0.07 | 0.3 | 0.21 | 0.56 | **0.21** |
| CMORPH | 5.91 | 0.12 | 0.46 | 0.43 | 0.38 | **0.21** |
| PERSIANN–CDR | **5.47** | −0.03 | 0.44 | 0.43 | 0.45 | 0.23 |

* The bold symbols present optimal performance.

The CC value represents the degree of linear correlation between satellite products and CN05.1. It was found that IMERG–F had the highest correlation (0.50) compared to the others. The KGE of IMERG–F was the highest and that of GSMaP was the lowest, which also indicated that IMERG–F was the closest to CN05.1 on the daily scale. As is shown in Table 3, IMERG products had a significantly higher POD (0.74) than others (<0.6), demonstrating that IMERG products could detect the precipitation events more accurately. FAR gives the fraction of the detected events for which the event had not occurred. A low FAR value means that the estimate had a low ratio of false alarms. Although the difference in FAR among the six precipitation products was small, GSMaP and CMORPH presented a lower false alarm percentage. In summary, the IMERG precipitation product had relatively higher estimated accuracies of ME, CC, and KGE than the other five products (IMERG–E, IMERG–L, GSMaP, CMORPH, and PERSIANN–CDR) in the daily scale precipitation evaluation.

Figure 2 shows the spatial distributions of daily mean precipitation for six precipitation products over mainland China from 2001 to 2015. All satellite precipitation products monitored the spatial pattern of precipitation in mainland China, gradually decreasing from the southeast to the northwest in China. GSMaP overestimated the precipitation in northwest China, while it underestimated the precipitation in southeast China. Notably, GSMaP had a more significant underestimation than CMORPH in southeast China. Except for GSMaP and CMORPH, other products had good spatial consistency with CN05.1. Furthermore, CMORPH performed worse than other products for rain bands of 1–2 and 3–5 mm/day. It is worth noting that all satellite products in the southeast of the Tibet Autonomous Region show the characteristics of precipitation, which CN05.1 does not show.

Figure 3 shows the frequency distribution of daily precipitation for ground interpolation product and six satellite precipitation products over mainland China from 2001 to 2015. The frequency was obtained by ratios between precipitation days with a certain range of amounts and total days (both precipitation days and no precipitation days). The highest precipitation frequency occurred in the weak precipitation intensity, lower than 4 mm/day, and the lowest precipitation frequency occurred in the strong intensity, more than 64 mm/day. During the range of 1–8 mm/day, the frequencies of the satellite precipitation products were lower than those of the ground interpolation product (i.e., CN05.1), except for PERSIANN–CDR, which was higher than the observations. After 8 mm/day, the satellite products were closer to the ground interpolation product. All but PERSIANN–CDR (25.9%) tended to underestimate the total number of precipitation events.

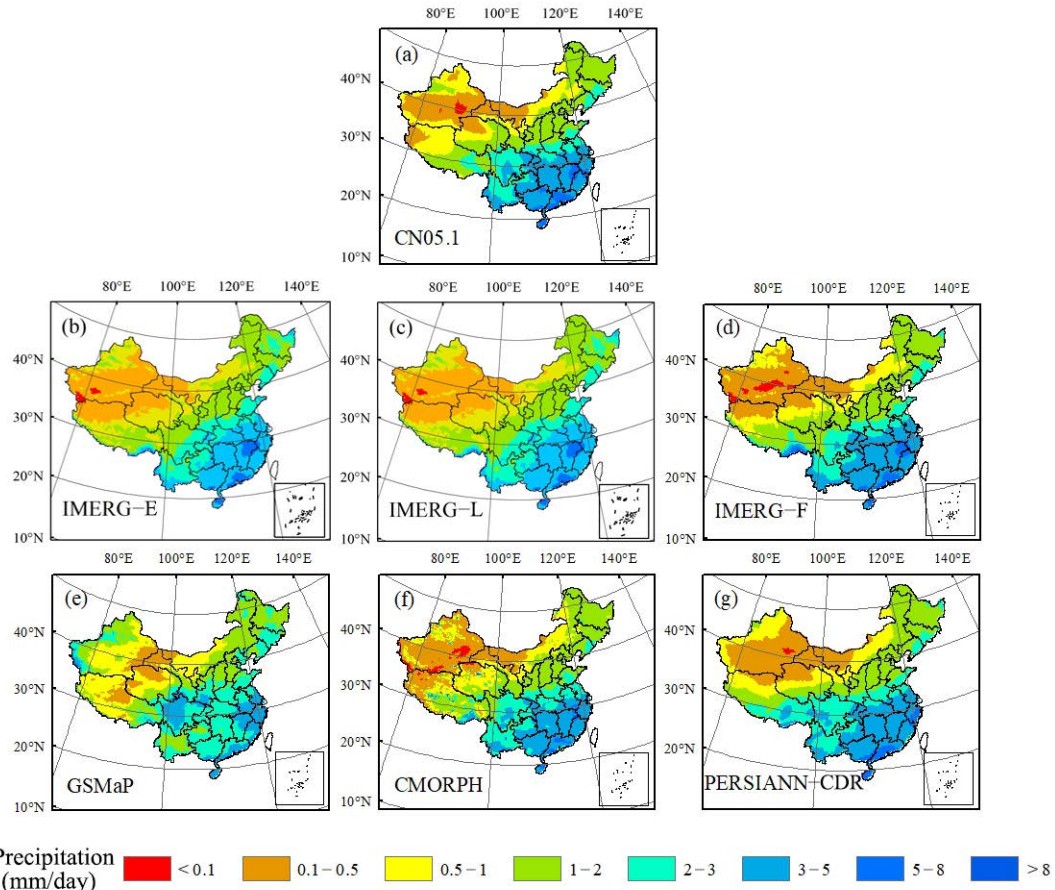

**Figure 2.** Distribution of daily mean precipitation over mainland China from 2001 to 2015.

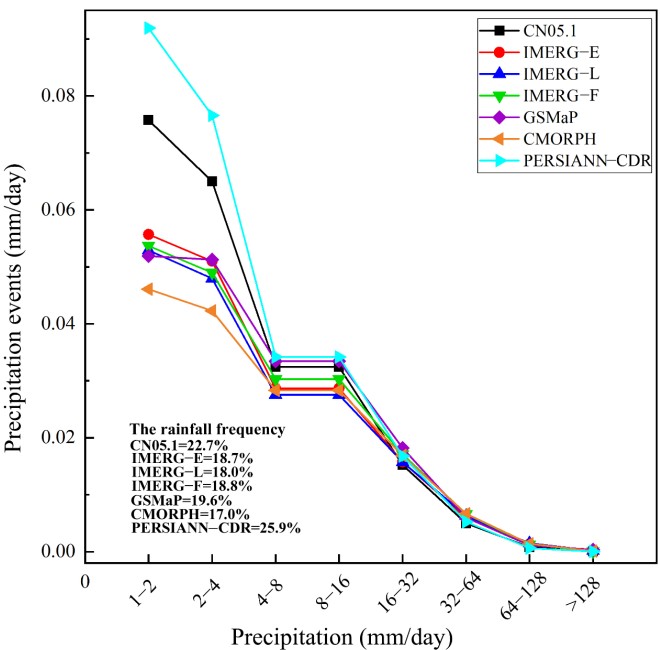

**Figure 3.** The frequency of daily precipitation from ground interpolation product (CN05.1) and six satellite products.

### 3.2. Monthly and Seasonal Precipitation

The scatter density plots of monthly precipitation between each satellite product and ground interpolation product of CN05.1 are shown in Figure 4. It can be seen that more grids in China presented a precipitation intensity lower than 2000 mm/month, and, thus, the high–density points were not easily found in Figure 4. In summary, the CC and KGE of monthly scale satellite precipitation were higher than those at the daily scale. Specifically, the range of monthly RMSEs was from 23.79 to 69 mm/month. The range of daily CC was from 0.30 to 0.50, and the range of monthly CC was from 0.63 to 0.95. The range of daily KGE was from 0.21 to 0.48, and the range of monthly KGE was from 0.48 to 0.94. In other words, the errors of the monthly scale satellite precipitation estimates were lower than those at the daily scale. This makes sense.

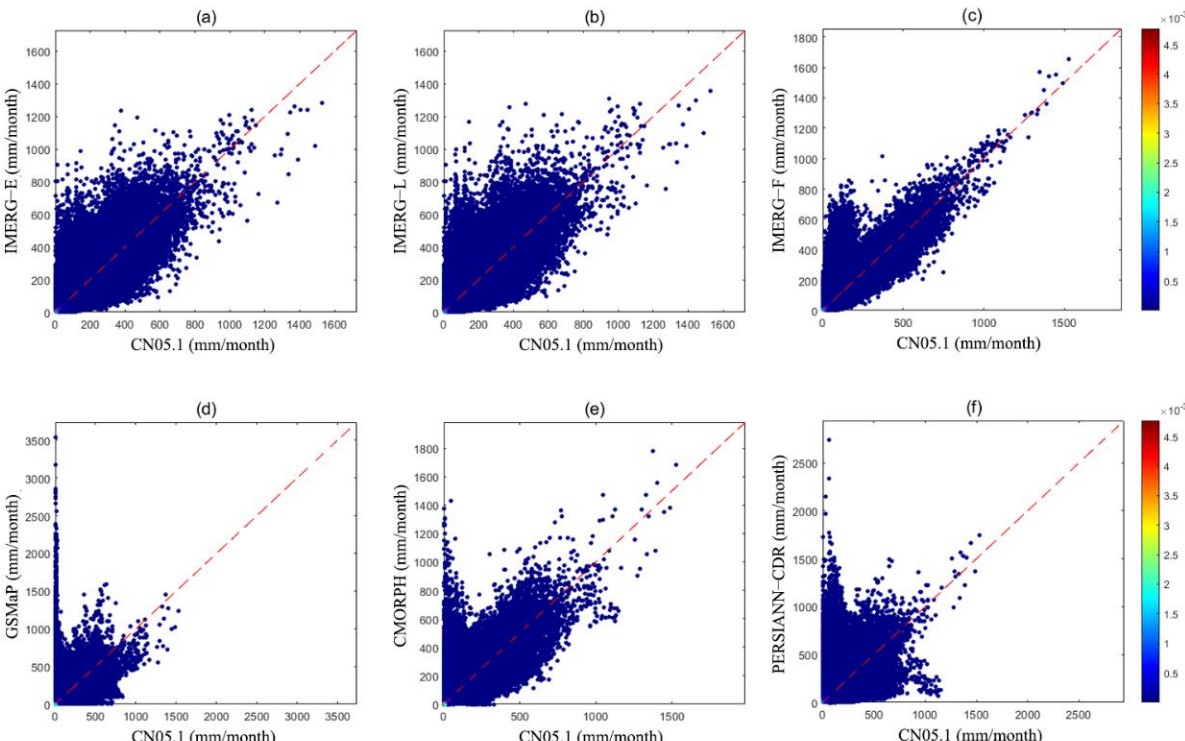

**Figure 4.** Scatter density plots of monthly precipitation for ground interpolation product (CN05.1) versus six satellite estimates: (**a**) IMERG–E, (**b**) IMERG–L, (**c**) IMERG–F, (**d**) GSMaP, (**e**) CMORPH, and (**f**) PERSIANN–CDR.

It can also be found that IMERG–F had the highest CC (0.95), the highest KGE (0.94), and the lowest RMSE (23.79 mm/month) among all six satellite products. GSMaP and PERSIANN–CDR had poor performance; their RMSEs were larger than 60 mm/month, and others were lower than 40 mm/month. Additionally, GSMaP had the lowest CC (0.63). In addition to RMSE, CC, and KGE, the ME of the six satellite precipitation products are shown in Table 4. IMERG–F had the lowest ME (0.31). In total, IMERG–F had the best monthly precipitation performance (i.e., the lowest RMSE and ME and the highest CC and KGE). This result is also consistent with the daily scale performance (see Table 3). The computation for the metric of POD and FAR at the monthly or yearly scale is meaningless, and therefore the POD and FAR were not demonstrated in the monthly and following yearly statistics. It is worth noting that, compared with IMERG–E and IMERG–L, the proximity between IMERG–F and CN05.1 after site correction was greatly improved (Figure 5a–c).

**Table 4.** Monthly precipitation statistics of the six satellite precipitation products over mainland China from 2001 to 2015 *.

|  | RMSE (mm/Month) | ME (mm/Month) | CC | KGE |
|---|---|---|---|---|
| IMERG–E | 37.32 | 2.2 | 0.87 | 0.86 |
| IMERG–L | 38.17 | 2.12 | 0.86 | 0.85 |
| IMERG–F | **23.79** | **0.31** | **0.95** | **0.94** |
| GSMaP | 60.85 | −2.1 | 0.63 | 0.63 |
| CMORPH | 35.9 | 3.52 | 0.88 | 0.86 |
| PERSIANN–CDR | 69 | −16.42 | 0.68 | 0.48 |

* The bold symbols present optimal performance.

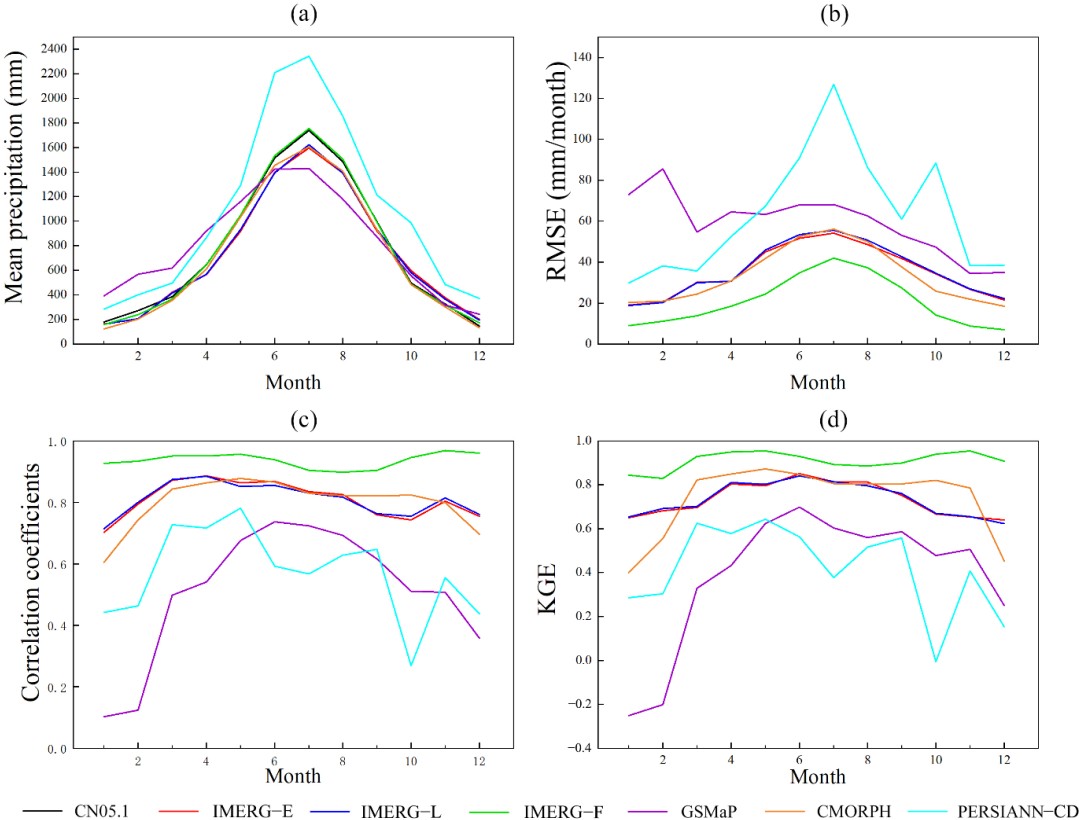

**Figure 5.** Monthly variations in statistics. (**a**) Mean precipitation (mm); (**b**) RMSE (mm/month); (**c**) CC; and (**d**) KGE.

The precipitation statistics in each month are shown in Figure 5. In terms of average precipitation, all satellite products captured significant seasonal variations (Figure 5a), with more precipitation in summer (June to August) and less precipitation in winter (December to February). The precipitation peak occurred in the period from June to August, which is consistent with the climatic characteristics of China in the East Asian monsoon region. The monthly precipitation variations of IMERG–F had a high degree of coincidence with CN05.1. Except for summer, IMERG–E, IMERG–L, and CMORPH also coincided with CN05.1 in other months. A whole overestimation occurred in PERSIANN–CDR, and the remaining products were underestimated in summer and basically overestimated in winter. As shown in Figure 5b, the RMSEs of all products except GSMaP at the monthly scale are lower in winter and higher in summer. Combined with Figure 5a, it can be concluded that the accuracy of precipitation inversion decreases as precipitation increases. The comparisons of monthly correlations are shown in Figure 4c. The monthly CC for GSMaP showed significant seasonal fluctuations, with high values in summer and low values in winter, but

the opposite is true for PERSIANN–CDR. However, the CC for other products presented relatively stable monthly variation. Notably, IMERG–F had the highest correlation from January to December and was followed by IMERG–E, IMERG–L, and CMORPH. KGE is similar to CC (see Figure 5).

Figure 5a shows that precipitation in mainland China has a significant seasonal characteristic regardless of satellite estimates and ground interpolation product. Therefore, the spatial distributions of seasonal precipitation for the different satellite products were compared to analyze the difference in satellite estimates at the seasonal scale. The spatial distributions of precipitation during the four seasons for ground interpolation product and the six satellite precipitation products are shown in Figure 6. Overall, Figure 6 shows that mainland China has a typical East Asian monsoon climate, which is characterized by more precipitation in summer and autumn and less precipitation in spring and winter. Spatially, there was more precipitation in the southeast near the sea and less precipitation in the northwest inland.

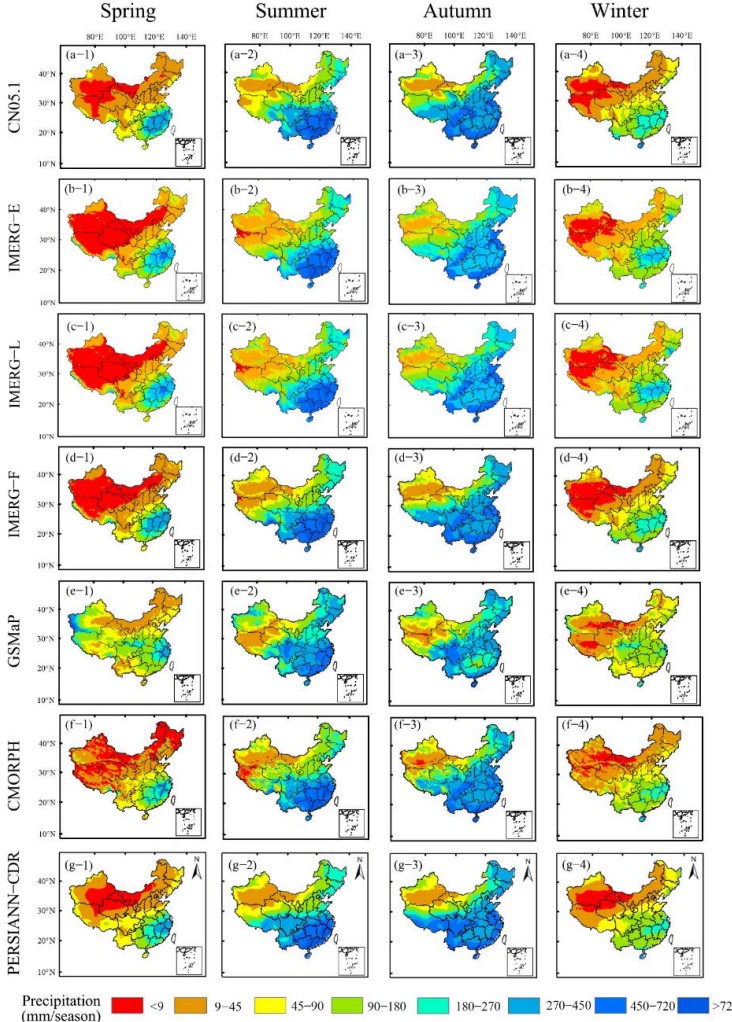

**Figure 6.** Comparisons of spatial distributions of seasonal average precipitation (mm/day) between six satellite products and ground interpolation product (CN05.1): (**a–1**)–(**a–4**) represent spatial distributions of mean precipitation for CN05.1 from spring to winter, respectively; (**b–1**)–(**b–4**) represent four seasons for IMERG–E; (**c–1**)–(**c–4**) represent four seasons for IMERG–L; (**d–1**)–(**d–4**) represent four seasons for IMERG–F; (**e–1**)–(**e–4**) represent four seasons for GSMaP; (**f–1**)–(**f–4**) represent four seasons for CMORPH; and (**g–1**)–(**g–4**) represent four seasons for PERSIANN–CDR.

Specifically, the different satellite products presented different precipitation characteristics. Compared with the ground interpolation product of CN05.1, the estimates of

IMERG products were drier in spring and winter for the Tibetan Plateau and northwest region in China. This may be one of the main errors for the IMERG products. In addition, IMERG products were wetter in southeast China for both summer and autumn. The IMERG products were wetter in spring and winter for the northeast region (see Figure 6(b–1~d–4)). GSMaP was drier than CN05.1 in humid southern China in all four seasons, which may be its main error source. In spring, the Tibetan Plateau and northwest region in China became wet, which is also one of the main differences between GSMaP and CN05.1 (see Figure 6(e–1~e–4)).

Compared with GSMaP and PERSIANN–CDR, the remaining IMERG products, CMORPH were closer to the ground interpolation product of CN05.1, as a whole (see Figure 6 and Table 5). Furthermore, CMORPH was slightly drier than CN05.1 for northeastern China and southeastern China in spring and winter, and CMORPH significantly underestimated precipitation in northeastern China in spring (see Figure 6(f–1)); however, there was no significant difference in summer and autumn. Different from PERSIANN–CDR, IMERG products and CMORPH presented better similarity with CN05.1. However, some slight differences also existed in areas with little or no precipitation(<0.1 mm/day) for IMERG products and PERSIANN–CDR. For example, under the prediction of IMERG products, the area of drought in the northwest region in spring is larger than the CN05.1 interpolation product. Specifically, IMERG products completely covered these observed regions, and PERSIANN–CDR covered part of the regions (see Figure 6).

**Table 5.** Seasonal precipitation statistics of the six satellite precipitation products over mainland China from 2001 to 2015 *.

| Season | Data | RMSE (mm/Season) | ME (mm/Season) | CC | KGE |
|---|---|---|---|---|---|
| Spring | IMERG–E | 47.26 | **1.93** | 0.88 | 0.82 |
| | IMERG–L | 47.15 | 2.13 | 0.88 | 0.82 |
| | IMERG–F | **27.57** | 4.83 | **0.95** | **0.89** |
| | GSMaP | 161.56 | −48.06 | 0.25 | 0.02 |
| | CMORPH | 50.77 | 10.12 | 0.82 | 0.68 |
| | PERSIANN–CDR | 75.78 | −25.28 | 0.67 | 0.54 |
| Summer | IMERG–E | 93.49 | 19.32 | 0.91 | 0.87 |
| | IMERG–L | 96.72 | 18.39 | 0.90 | 0.86 |
| | IMERG–F | **63.92** | **−0.84** | **0.96** | **0.95** |
| | GSMaP | 147.27 | −18.68 | 0.75 | 0.67 |
| | CMORPH | 94.51 | 7.57 | 0.91 | 0.90 |
| | PERSIANN–CDR | 163.53 | −20.39 | 0.77 | 0.72 |
| Autumn | IMERG–E | 106.32 | 17.61 | 0.86 | 0.84 |
| | IMERG–L | 110.96 | 14.99 | 0.85 | 0.84 |
| | IMERG–F | **92.63** | **−2.25** | **0.90** | **0.89** |
| | GSMaP | 145.47 | 48.78 | 0.74 | 0.60 |
| | CMORPH | 111.16 | 18.37 | 0.86 | 0.84 |
| | PERSIANN–CDR | 223.64 | −91.91 | 0.64 | 0.51 |
| Winter | IMERG–E | 59.85 | −12.47 | 0.80 | 0.70 |
| | IMERG–L | 59.86 | −10.04 | 0.81 | 0.70 |
| | IMERG–F | **21.31** | **2.03** | **0.97** | **0.95** |
| | GSMaP | 81.08 | −7.30 | 0.52 | 0.51 |
| | CMORPH | 47.65 | 6.20 | 0.83 | 0.79 |
| | PERSIANN–CDR | 125.64 | −59.51 | 0.36 | 0.17 |

* The bold symbols present optimal performance.

The seasonal scale statistical indicators of each product are shown in Figure 5. The RMSE of IMERG–F is the lowest among all products in all seasons, and CC and KGE are also better than other products, which indicates that IMERG–F has the most accurate precipitation inversion on the seasonal scale. It is worth noting that the ME of IMERG–E is smaller than that of IMERG–F in spring.

### 3.3. Annual Precipitation

The scatter density plots of annual precipitation for the satellite product versus CN05.1 are shown in Figure 7. The range of yearly RMSEs was from 166.19 to 361.22 mm/year, and the range of yearly CCs was from 0.72 to 0.95. The best performance still occurred in the IMERG–F product estimate, which had the lowest RMSE (166.19 mm/year) and highest CC (0.95) among the six satellite products. It can be concluded that the IMERG–F estimate was optimal among the six satellite products at daily, monthly, and annual scales. In contrast, GSMaP had the worst RMSE (361.22 mm/year), lowest correlation (0.72), and KGE (0.63). Further comparisons are shown in Table 6. Notably, IMERG–F also had the smallest bias (3.79 mm/year). Compared with the respective monthly correlation, the annual correlation of GSMaP and CMORPH significantly improved, varying from 0.63 to 0.72 for GSMaP and from 0.88 to 0.91 for CMORPH; the annual correlation of IMERG–F basically remained the same; and the annual correlation of PERSIANN–CDR significantly increased from 0.72 to 0.94 (see Table 6).

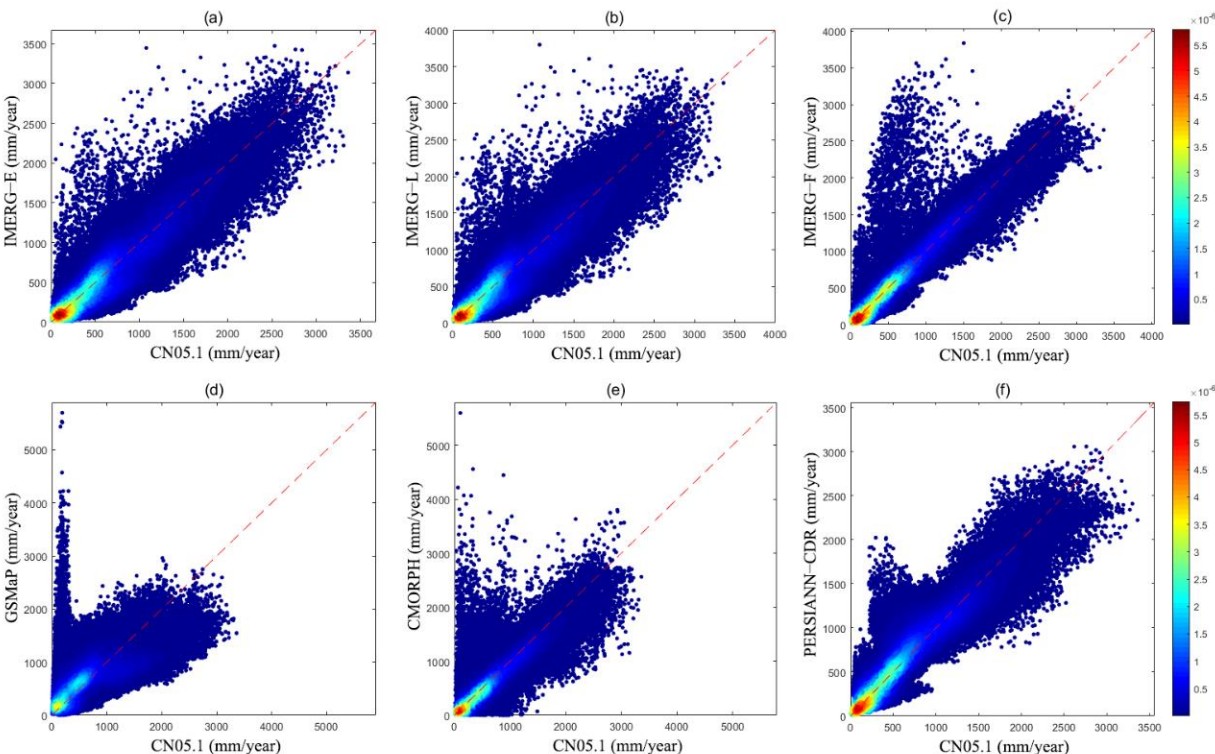

**Figure 7.** Scatter density plots of annual precipitation for ground interpolation product (CN05.1) versus six satellite estimates: (**a**) IMERG–E, (**b**) IMERG–L, (**c**) IMERG–F, (**d**) GSMaP, (**e**) CMORPH, and (**f**) PERSIANN–CDR.

**Table 6.** Annual precipitation statistics of the six satellite precipitation products over mainland China from 2001 to 2015 *.

|  | RMSE (mm/Year) | ME (mm/Year) | CC | KGE |
|---|---|---|---|---|
| IMERG–E | 201.5 | 26.39 | 0.92 | 0.91 |
| IMERG–L | 211.28 | 25.47 | 0.92 | 0.91 |
| IMERG–F | **166.19** | **3.79** | **0.95** | **0.94** |
| GSMaP | 361.22 | −25.63 | 0.72 | 0.63 |
| CMORPH | 219.97 | 42.22 | 0.91 | 0.88 |
| PERSIANN–CDR | 171.77 | −10.17 | 0.94 | 0.93 |

* The bold symbols present optimal performance.

The spatial distributions of annual precipitation for ground interpolation product and six satellite products are shown in Figure 8. As a whole, GSMaP had significant spatial differences with ground interpolation product of CN05.1, and its CCs was lower than 0.9 (see Table 6); the remaining products, including IMERG products, CMORPTH, and PERSIANN–CDR, demonstrated higher spatial consistency with CCs higher than 0.9.

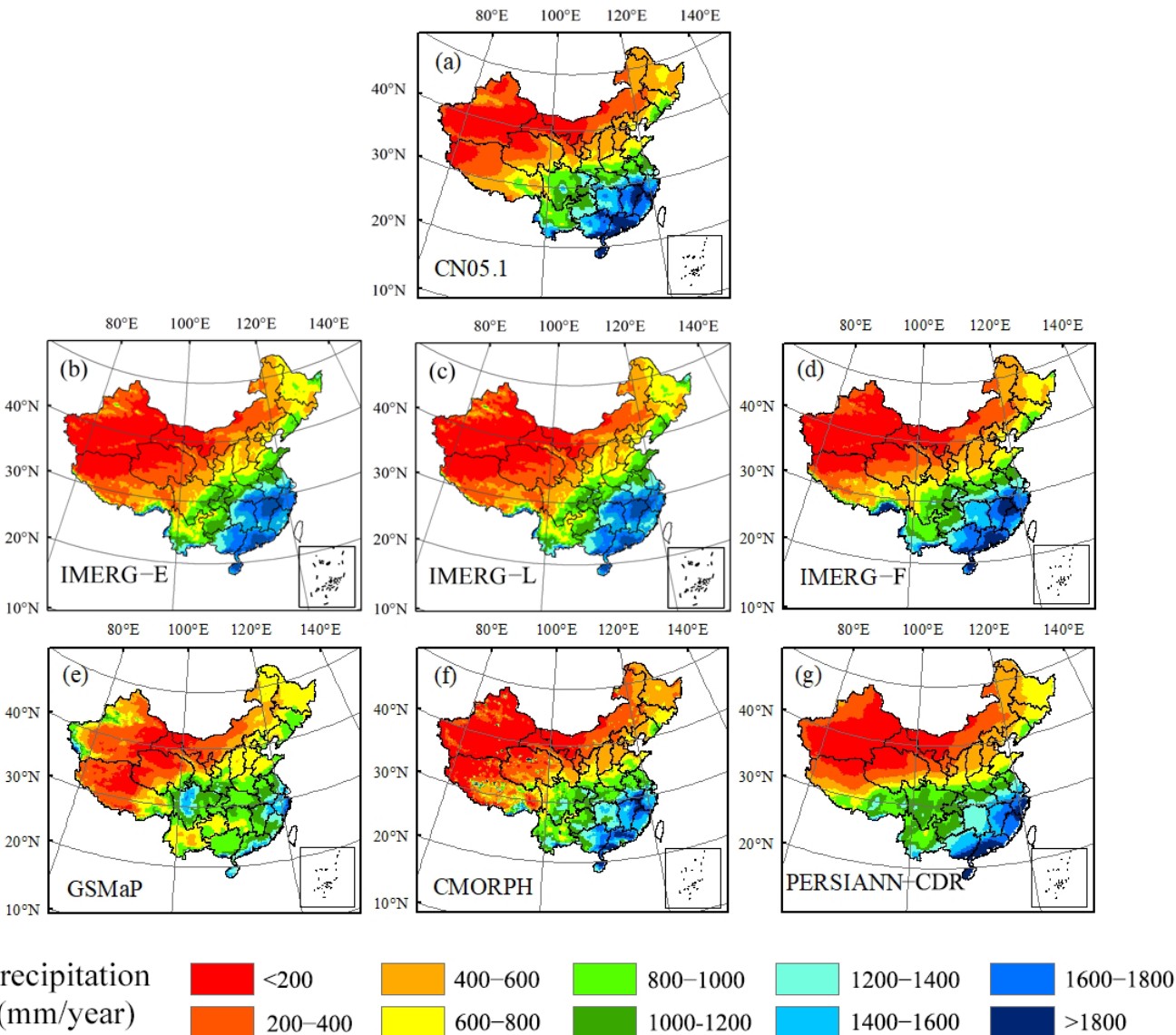

**Figure 8.** Comparisons of spatial distributions of annual precipitation between six satellite products and ground interpolation product (CN05.1): (**a**) CN05.1, (**b**) IMERG–E, (**c**) IMERG–L, (**d**) IMERG–F, (**e**) GSMaP, (**f**) CMORPH, and (**g**) PERSIANN–CDR.

Specifically, PERSIANN–CDR was significantly wetter than CN05.1 on the southeast of the Qinghai–Tibet Plateau. Except for CMORPH, all the other products underestimated the precipitation in Northeast China. GSMaP was significantly drier than CN05.1 in the humid southeastern and southwestern regions of China. Conversely, it was slightly wetter than the CN05.1 in northern China. However, CMORPH still had some significant differences in local regions compared with others. For instance, CMORPH was relatively drier in southeastern and northeastern China than IMERG products and PERIANN–CDR. Visually, there were no significant spatial differences in precipitation between IMERG products and PERIANN–CDR, but based on Table 6, IMERG products performed better than PERIANN–

CDR (lower RMSE, higher CC and KGE). Therefore, IMERG products were quantified as the optimal estimate of annual precipitation over mainland China from 2001 to 2015.

## 4. Discussion

This study analyzed the reliability and quality of six readily available gridded satellite precipitation products compared to ground interpolation product. The result indicated that IMERG–F was the overall best performing precipitation product in China, followed by IMERG–E, IMERG–L, PERSIANN–CDR, CMORPH, and GSMaP, which is consistent with the conclusion that higher spatial–temporal precipitation products were suggested to be of better quality compared to the lower–resolution product [56]. However, the performance of GSMaP was not ideal, which may be related to its poor accuracy in processing raw data and calibrated site data JRA–55 using only the cloud movement vector method and Kalman filter method. This indicates that the performances of satellite precipitation products could be affected by various factors, such as input data, onboard sensors, algorithms, and interpolation techniques [51]. IMERG products were significantly more consistent with CN05.1 in winter (Figure 6). The reason may be that IMERG's precipitation radar (PR) adds Ka–band radar, which is more sensitive and can observe ice grains, therefore increasing the ability of GPM to detect solid precipitation [47]. In addition, IMERG–F used gauged GPCC to correct the basis of IMERG–E and IMERG–L, and therefore, the former is better than the latter at all scales [56].

In terms of satellite precipitation products, all the satellite products in this study accurately show the difference between the precipitation systems in southeast and northwest China. Among all the products, the inversion of precipitation by IMERG products and PERSIANN–CDR accurately delineates the 800 mm and 400 mm isohyets of China. Although CMORPH also roughly delineates the 800 mm isohyets of the Qinling–Huaihe River, the inversion of precipitation in the central and northwest regions is not continuous. There were high values that were not in line with the actual situation. Many other studies found that satellite precipitation products overestimate the precipitation in arid and semiarid regions [57,58]; this phenomenon is also reflected in this study (see Figure 2). The more convincing explanation is that trace amounts of precipitation in clouds were measured by satellite but evaporated before landing to the ground, which is a normal phenomenon for arid and semiarid areas [59]. The Qinghai–Tibet Plateau in southwest China has harsh natural conditions and few observation stations. However, the southeastern part of the region is influenced by warm and humid air currents from the Indian Ocean, which may result in more precipitation than CN05.1 The precipitation retrieved by satellite may be closer to the actual situation. The key assumption here is that satellite depictions of precipitation outperform interpolation methods in areas where few data are available [60].

The precipitation prediction of each product showed obvious seasonal fluctuations, which is consistent with the climatic characteristics of China. However, in months with more precipitation, RMSE was higher and CC was lower, which indicated that the more precipitation there is, the larger the error of precipitation inversion. The seasonal variation in POD and FAR was more obvious, which is consistent with previous research results [61,62]. The reason for the low POD in winter may be that the satellite algorithm cannot accurately obtain the weak precipitation signal [63]. The obvious difference in the detection capabilities of precipitation in winter and summer (or cold and warm seasons) is a challenge facing all satellite products [64]. The detection ability of precipitation in winter is weaker than that in summer, a circumstance which may be the cause of cold snow cover being mistaken for precipitation in complex underlying surfaces [65,66]. Other studies also mentioned that satellite precipitation detection is better in summer than in winter over mainland China [63]. In general, the poor performance of satellite precipitation in winter may be caused by the cold ground affecting the inversion of PMW, or the satellite's ability to detect low–intensity precipitation may be limited [64].

Although the observation of CN05.1 has covered more information of ground precipitation stations, the data at point scale are limited to represent the actual regional scale

information, especially in northwest China with sparse stations. In other words, the site observation has greater uncertainty to represent regional information; however, triple collocation analysis (TCA) is a comprehensive validation method independent of the site observations. It estimates the random error variance and signal–to–noise ratio of satellite precipitation products using their independence and linear correlation with the known "truth value" [67,68]. Its main advantages include the following: (1) it is independent of the site observation and suitable for the evaluation on the areas without site observation; (2) it reduces the uncertainty of error estimation caused by different spatial resolution between evaluated data and observed data. Therefore, the TCA method will be used to obtain a more reasonable and robust evaluation for satellite precipitation products in future study.

## 5. Conclusions

The spatial heterogeneity of the spatial distribution of precipitation and the insufficient density of rain gauges seriously affect the observation of precipitation. The ability of satellite precipitation products to provide continuous precipitation data also makes them an attractive option. Numerous verifications of satellite–estimated precipitation have been performed in the past, but few studies have focused on China using a relatively long data record. In this study, six satellite precipitation products were evaluated by comparison with a gauge–based interpolation of daily precipitation data over mainland China during a fifteen-year period from 2001–2015. The findings of this study included the following:

1. For the correlation coefficient (CC), the fluctuation range of each product on the daily scale was 0.3–0.5, the monthly scale was 0.63–0.95, and the annual scale was 0.72–0.95. The correlation between satellite products and CN05.1 was significantly improved from day to year. The CC of IMERG–F was generally better than that of IMERG–E and IMERG–F, which showed that the station correction can significantly improve the accuracy of satellite precipitation products. IMERG–F was only slightly worse than PERSIANN–CDR at the daily scale, but its RMSE, ME, and CC were all optimal at the other scales. Especially at the monthly scale, the error ME of IMERG was almost 0. The high POD and FAR on the daily scale indicated that it may improve the hit rate of precipitation events through a large number of forecasts but at the same time make the false alarm rate higher.

2. In terms of spatial distribution, all products can trace the precipitation difference between southeast and northwest China. IMERG products and PERSIANN–CDR can even better depict the precipitation lines in China. GSMaP overestimated the precipitation in western and northwestern China and even overestimated it in spring and winter across mainland China. In spring and winter, IMERG products significantly underestimated the precipitation in northwest China. It is worth noting that there are few observation stations in the Qinghai–Tibet Plateau, and the inversion of precipitation by satellite products may be closer to the actual local precipitation.

3. CN05.1 site interpolation data showed that the probability of precipitation in mainland China was 22.7%, and the closest was GSMaP (19.6%), followed by PERSIANN–CDR (25.9%). This study also found that most of the mean annual precipitation was less than 16 mm/day. However, within the range of 1–16 mm/day precipitation, only PERSIANN–CDR was more than CN05.1, and other products were 0–25% less than CN05.1. The underestimation of precipitation less than 16 mm/day was also reflected in the monthly and annual scatter plots. All products were consistent with the trend of CN05.1 in the range of 4–128 mm/day, and almost all products had strong prediction ability for moderate and heavy precipitation.

4. All satellite precipitation products can well capture the seasonal variation in precipitation. The monthly precipitation of IMERG products and CMORPH was in good agreement with CN05.1. while PERSIANN–CDR showed overestimation of precipitation and GSMaP overestimation in the dry season and underestimation in the rainy season.

5. This study found that IMERG products (in particular IMERG–F) and PERSIANN–CDR have good spatial and temporal coincidence with CN05.1, which can be applied to many fields, such as hydrology, meteorology, and disaster prediction in the Chinese mainland. Second, CMORPH also showed good performance in southeast, northeast, and East China.

Even the most complex techniques, such as radar, have difficulty quantifying precipitation values because of the high thermodynamic and kinetic complexity of precipitation, resulting in robust parametric processes [69]. Therefore, it will be feasible to develop and select a suitable algorithm for the fusion of multisource precipitation products. Some researchers have made promising progress in this regard [70,71], but more research is needed in the future.

**Author Contributions:** Conceptualization, Z.L. and Z.D.; methodology, Z.L.; validation, Z.L., Z.D., P.Q. and S.Z.; data curation, S.Z. and Q.M.; writing—original draft preparation, Z.L.; writing—review and editing, Z.L. and Z.D.; visualization, Z.L.; supervision, Z.D.; funding acquisition, P.Q. and Q.M. All authors have read and agreed to the published version of the manuscript.

**Funding:** This research was supported by the National Key Research and Development Program of China (Grant No. 2019YFA0606903), the National Natural Science Foundation of China (Grant No. 41930970 and 41571368), and the National Key Scientific and Technological Infrastructure project "Earth System Science Numerical Simulator Facility" (EarthLab).

**Data Availability Statement:** The precipitation datasets used in our work can be freely accessed at the following websites: IMERG–E, IMERG–L, and IMERG–F: https://disc.gsfc.nasa.gov/ (accessed on 21 August 2022); GSMaP: https://sharaku.eorc.jaxa.jp/GSMaP/index.htm (accessed on 30 October 2021); CMORPH: https://rda.ucar.edu/ (accessed on 15 October 2021); PERSIANN–CDR: http://chrsdata.eng.uci.edu/ (accessed on 3 August 2022).

**Conflicts of Interest:** The authors declare no conflict of interest.

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
