# Peer review of "Evaluation of Six Satellite Precipitation Products over the Chinese Mainland"

_remotesensing, doi:10.3390/rs14246277_

Round 1

Reviewer 1 Report (Previous Reviewer 1)

The authors address all the comments provided for the previous version

Reviewer 2 Report (Previous Reviewer 2)

Accept 

Reviewer 3 Report (Previous Reviewer 3)

Thank you for addressing my comments and reviewing the manuscript accordingly. I have no further comments and recommend to accept the revised manuscript in present form. 

This manuscript is a resubmission of an earlier submission. The following is a list of the peer review reports and author responses from that submission.

Round 1

Reviewer 1 Report

Summary:

This manuscript validates the satellite based precipitation products using ground-based gauge observations over mainland chine. The authors validates six different precipitation products with different resolution and latencies. 

It would have been more helpful to validate the current operational products that the discontinued product. Thus, the manuscript lacks significance. I would recommend the authors to redo the analysis for currents products and resubmit. 

Here are my comments:

1. TRMM products are now discontinued and superseded by IMERG. What is significance of validating TRMM products? 

2. As the authors mentioned, there are three products associated with IMERG. Not sure if I missed it, but which product has been evaluated here? Instead of evaluating two version of TRMM (which is not active anymore), it would make more sense to validate all three products of IMERG. 

3. IMERG and GSMaP has higher spatial resolution than other satellite products and ground-data used here. How is this difference in spatial resolution handled? 

4. Figure 4, 7: The colorbar is not very helpful. Maybe changing the scale would help. 

Reviewer 2 Report

Review for “Evaluation of Six Satellite Precipitation Products over the Chinese Mainland”

The review process was very difficult as the line numbering was missing in the manuscript. However, it can be accepted after some major comments. The manuscript is generally good and the topic seems to present very interesting results for readers. I suggest a "minor" revision before possible consideration of the application in Remote sensing. My comments are listed as:

1.    Although the manuscript is generally well-written, a language check by a professional native speaker or an editing agency is needed to fix some syntax, style, and phrasing problems.

2.    Please check reference style as all references had wrong style (add a space between the word and the reference).

3.    Why the authors used study period till 2015, while GSMaP is available more than 2015 as they claimed. Check “Evaluating intensity-duration-frequency (IDF) curves of satellite-based precipitation datasets in Peninsular Malaysia”

4.    The introduction needs to be improved and further discussion is needed. Also, please use either rainfall or precipitation in the manuscript. Also, introduction required adding newly references.

5.    Figure 1: add elevation map and some features to the map.

6.    Add any table or figure directly after their first mention.

7.    Many GSMaP products written in page 5. Please identify each one of them.

8.    Check sentence start with “FAR gives the…..” in section 2.3

9.    Use KGE metric as it can evaluate multiple properties together in one integrated metric. It combines Pearson's correlation (r), the ratio of spatial variability and the normalized difference.

10. Figure 2: increase the size of the figure, add legend title and units, and check caption.

11. Figure 5, 6: increase size of the figure

12. Figure 4 and 7: increase size of the figure, add only 1 legend with title and units

13. Page 12 last paragraph: You wrote “closer to” in 2nd line. I would like to know how much it closer to?

14. All figures should have legend title with units.

15. Please check the references as there are many references without Doi. Please check them carefully.

I look forward to seeing a better version of the manuscript.

Reviewer 3 Report

Dear Editor,

Please find my review of a study " Evaluation of Six Satellite Precipitation Products over the Chinese Mainland" by Zhenwei Liu , Zhenhua Di , Peihua Qin , Shenglei Zhang , Qian Ma submitted to Remote Sensing, Section: Atmospheric Remote Sensing, Special Issue: Remote Sensing of Precipitation: Part III for consideration for possible publication.

This study aims to evaluate accuracy of satellite precipitation datasets including TRMM-3B42V7, TRMM-3B42RT, IMERG, GSMaP, CMORPH and PERSIANN-CDR at different temporal scales; study area is mainland China, study period is from 2001 to 2015. Based on obtained results, the authors draw conclusions about performance of the evaluated   satellite precipitation datasets.

The subject of this report is suitable for Remote Sensing and it could be published after suggested revision.

Abstract

Delete " In general, this study analyzed the error characteristics of six satellite precipitation products in mainland China." This sentence is redundant, you already mentioned it earlier.

(3) The performance of six kinds of satellite precipitation products in summer is better than that in winter. => (3) The performance of six types of satellite precipitation products in summer was better than that in winter.

However, the error will be larger in seasons with more precipitation. => However, the error was larger in seasons with more precipitation.

1. Introduction

Page 2. Precipitation is one of the important meteorological variables in research fields such as drought assessment and detection . please add references

Page 2. and its accurate measurement helps to conduct more reasonable studies of hydrological and meteorological events. => and its accurate measurement helps to conduct more comprehensive studies of hydrological and meteorological events.

Page 2. precipitation has large variability in temporal and spatial distribution => precipitation has large variability in temporal and spatial domains

Page 2. The traditional methods measuring rainfall mainly include rain gauges and groundbased radar => The traditional methods of measuring rainfall mainly include rain gauges and ground-based radars.

Page 2. especially for nondata areas => especially for limited data areas

Page 2. The following paragraph should be deleted as it presents trivial information known to the readers. " Groundbased radar can reflect the spatial heterogeneity of precipitation well.  It transmits electromagnetic waves through the radar antenna. If the wave encounters clouds, rain, snow and other objects, part of the radiation energy is returned and received by the radar antenna, forming a meteorological echo. The energy difference between sending and receiving waves was retrieved from three-dimensional precipitation characteristics such as strength, volume and start-stop time data."

Page 3. Recently, a study on evaluating Satellite Precipitation Estimates over Australia has been published in Remote Sensing. 2022, 14, 2724. https://doi.org/10.3390/rs14112724. Suggest including it in Introduction, first paragraph.

2. Materials and Methods

Page 3. It absorbed more than 2,400 national stations => It included more than 2,400 national stations

Page 3. Climate Prediction Center morphing technique (CMORPH) => NOAA's Climate Prediction Center morphing technique (CMORPH)

Page 3. estimates from various passive microwave (PMW) sensors, => estimates from various PMW sensors; acronym has been introduced on Page 2.

Page 5. CMORPH is a new technology developed by the Climate Prediction Center (CPC) of the United States => CMORPH is a new technology developed by the Climate Prediction Center (CPC) of the United States National Oceanic and Atmospheric Administration (NOAA).

Page 5. The primary characteristic is the integration of the infrared data of geostationary satellites and microwave data of polar satellites for the estimation of precipitation. => The primary characteristic is the integration of the IR data of geostationary satellites and MW data of polar satellites for the estimation of precipitation.

Please make similar corrections to IR and MW in the following sentences of this paragraph.

Page 6. The brief formula descriptions are given as follows. => ME, RMSE, CC, POD and FAR are calculated using Equations 1 – 5, respectively.

3. Results

Page 7. It was found from Table 3 => As one can see from Table 3

4. Discussion.

In this study, the CN05.1 dataset was used as the benchmark to evaluate the performance of different satellite products over Mainland China from 2001 to 2015.

On the other hand, some earlier studies used Triple Collocation Analysis (TCA) to evaluate accuracy of satellite precipitation estimates. Please discuss the TCA methodology and provide your explanations how it could be applicable to your study.

Page 18. Conflicts of Interest: The authors declare no conflict of interest. Delete the text from template which is not relevant to you.

Page 19. You have no Appendices – please delete.

This reviewer recommends accepting the manuscript for publication after a minor revision.

Yours faithfully,

The Reviewer